# Pointing Error Correction for Vehicle-Mounted Single-Photon Ranging Theodolite Using a Piecewise Linear Regression Model

**DOI:** 10.3390/s24103192

**Published:** 2024-05-17

**Authors:** Qingjia Gao, Chong Wang, Xiaoming Wang, Zhenyu Liu, Yanjun Liu, Qianglong Wang, Wenda Niu

**Affiliations:** Changchun Institute of Optics, Fine Mechanics and Physics, Chinese Academy of Sciences, Changchun 130033, China; gaoqj@ciomp.ac.cn (Q.G.); wangchong@ciomp.ac.cn (C.W.); liuyanjun@ciomp.ac.cn (Y.L.); wangqianglong@ciomp.ac.cn (Q.W.); niuwenda@ciomp.ac.cn (W.N.)

**Keywords:** single-photon ranging theodolite, pointing error, correction, piecewise linear regression

## Abstract

Pointing error is a critical performance metric for vehicle-mounted single-photon ranging theodolites (VSRTs). Achieving high-precision pointing through processing and adjustment can incur significant costs. In this study, we propose a cost-effective digital correction method based on a piecewise linear regression model to mitigate this issue. Firstly, we introduce the structure of a VSRT and conduct a comprehensive analysis of the factors influencing its pointing error. Subsequently, we develop a physically meaningful piecewise linear regression model that is both physically meaningful and capable of accurately estimating the pointing error. We then calculate and evaluate the regression equation to ensure its effectiveness. Finally, we successfully apply the proposed method to correct the pointing error. The efficacy of our approach has been substantiated through dynamic accuracy testing of a 450 mm optical aperture VSRT. The findings illustrate that our regression model diminishes the root mean square (RMS) value of VSRT’s pointing error from 17″ to below 5″. Following correction utilizing this regression model, the pointing error of VSRT can be notably enhanced to the arc-second precision level.

## 1. Introduction

A photoelectric theodolite collects flight target information through optical imaging and obtains the required target parameters by coordinate transformation, time and space registration, and intersection calculation. It is an essential component of a spacecraft launch and recovery measurement and control system, also widely utilized in target detection [1,2,3]. However, traditional photoelectric theodolites either lack ranging information or their ranging ability does not match the operating distance of the optical imaging system. Therefore, it becomes necessary to adopt a two-station or multi-station intersection measurement method [4]. With advancements in single-photon high-sensitivity detectors and high-precision timers, vehicle-mounted single-photon ranging theodolites (VSRTs) with high time resolution can detect distances of targets even under extremely low laser echo energy conditions [5,6]. A VSRT exhibits significant application prospects in single-station, high-precision real-time positioning measurements. Nevertheless, due to manufacturing and assembly errors, actual optical axial direction deviations from desired pointing occur in VSRTs; this is referred to as pointing error. Such errors impact target acquisition efficiency, tracking accuracy, imaging quality, and ranging precision. Although high-precision mechanical processing combined with frequent adjustments can enhance pointing accuracy, these options are expensive and time-consuming for correction purposes. To improve the overall performance of VSRTs, further research on error modeling and pointing correction technology is necessary.

The correction of pointing error is typically categorized into two types: hardware correction and digital correction. Hardware correction involves utilizing an external device to acquire the transformation matrix between angular deviation and coordinate elements, which is then compensated for in the control system [2,7,8,9]. On the other hand, digital correction employs error modeling and sample data to derive error parameters, enabling angle compensation for the swift calibration of an optical axis orientation. In comparison to hardware correction, digital correction reduces development costs while significantly enhancing pointing accuracy.

There are three primary methods for digitally correcting pointing error: the basic parameter method, the semi-parameter regression method, and the non-parameter regression analysis method. The basic parameter method involves establishing a pointing model that includes error parameters through ray tracing or homogeneous coordinate transformation. In this approach, each error parameter corresponds to a clear physical meaning, enabling the analysis of the influence of each error parameter on the direction of the optical axis. This can provide valuable guidance for the design and installation of optical machine systems in academic research and professional practice. Huang et al. utilized the basic parameter method to establish a mathematical model that describes the relationship between assembly error and pointing error in a spatial optical camera [10]. Zhou et al. employed a ray tracing algorithm to analyze the mechanical error of an aerial camera, subsequently establishing a pointing model. The identification of error parameters was accomplished through genetic algorithm optimization [11]. Peng et al. developed a linear model for calculating the pointing error of optical communication terminals on moving platforms, which consisted of an error parameter calculation model and a guiding value calculation model. The proposed model’s effectiveness was validated through tracking star experiments [12]. He et al. utilized a linear model to rectify the pointing error in the satellite–ground quantum experiment [13]. Similarly, Zhang et al. constructed a pointing error correction model for a space laser communication system through a geometric error analysis [4]. Furthermore, Yan et al. developed a linear model of pointing error based on the target positioning process and derived an optimized parameter model using the stepwise regression method, effectively reducing the azimuth error to 14″ and the elevation error to 12″ [14]. It can be seen that the basic parameter model has demonstrated significant efficacy in correcting pointing error. 

The semi-parametric regression method is capable of accommodating non-linear factors such as micro-vibration and structural deformation, thereby demonstrating commendable compensatory effects in scenarios involving substantial non-linear errors. The focus of research on semi-parametric regression lies in developing methodologies for estimating non-linear error regression models. Xu et al. utilized the wavelet denoising algorithm based on a soft threshold to estimate the nonlinear error and correct pointing error of the airborne photoelectric platform [15]. Liu and Huang, respectively, employed the compensated least square method to estimate and correct the nonlinear errors of aerial cameras and telescopes [16,17]. Additionally, Peng et al. applied the K-Nearest Neighbor Algorithm and kernel weight function to achieve successful correction of pointing errors in optical communication terminals [18,19]. The non-parametric regression analysis method does not necessitate intricate modeling. It primarily constructs error surfaces corresponding to the positional orientation of each optical axis through experimental testing and then fits them using spherical harmonic functions or a generalized extended approximate model to compensate for any pointing error [20,21]. Similarly, Xu et al. proposed a model based on a radial basis function neural network and conducted experiments at a mobile laser ranging station [22]. Tang et al. utilized Gaussian process regression to rectify the pointing error of the photoelectric detection system [23]. Both the semi-parametric and non-parametric models require a substantial amount of test data for validation.

The azimuth range of a VSRT is 0 to 360°, which requires pointing error correction to cover the entire operating range, unlike optical communication terminals, aerial cameras, and space optical cameras with limited azimuth. The distribution of pointing error varies in different quadrants and includes mechanical errors of the tracking frame itself as well as dynamic tracking errors, graphics processing errors, vehicle platform shaking errors, attitude sensor measurement errors, positioning and orientation instrument errors, and other sources for VSRTs [24,25]. Currently, there is a lack of reports addressing the correction of pointing error across the entire operational range of a VSRT, and a comprehensive investigation into the impacts of ranging and imaging systems on pointing error is needed. In this study, we propose a piecewise linear regression model based on kinematics models and optical image characteristics to effectively correct the pointing error of VSRT. 

## 2. Working Principle

### 2.1. Structure of VSRT

Figure 1 depicts the components of a VSRT, including a tracking frame with a horizon U-shaped structure comprising azimuth shafting, elevation shafting, optical shafting, driving motors, encoders, and supporting structures to facilitate azimuth and elevation motion of the single-photon ranging and optical imaging system. The optical imaging system is responsible for target acquisition, tracking, and imaging through its composition of the imaging lens group, focusing and magnifying mechanism, diaphragm, and detector. Furthermore, the single-photon ranging system encompasses the laser-transmitting and -receiving optical path for precise distance measurement of the target. Lastly, a vehicle supporting platform has been designed to provide a stable measurement base for the VSRT.

### 2.2. Correction Principle

For further correction, the pointing error is modeled using six right-handed coordinate systems, as depicted in Figure 1 and Figure 2. Ground plane coordinate system $G(x_{g}y_{g}z_{g})$: the origin is at the theodolite measuring station; $o_{g}x_{g}$ points northward; and $o_{g}z_{g}$ points towards the zenith. Basic platform coordinate system $B(x_{b}y_{b}z_{b})$: the origin is located at the center of the base plane, $o_{b}x_{b}$ points northward, and $o_{b}z_{b}$ points towards the zenith. Azimuth axis coordinate system $A(x_{a}y_{a}z_{a})$ and elevation axis coordinate system $E(x_{e}y_{e}z_{e})$: while $o_{a}y_{a}$ and $o_{e}y_{e}$ are parallel to the ideal pointing direction, the $o_{a}z_{a}$ and $o_{e}x_{e}$ axes are coincident with the azimuth axis and elevation axis, respectively. Line-of-sight (LOS) coordinate system $L(x_{l}y_{l}z_{l})$: its origin is at the intersection of the optical axis of the theodolite and tracking frame’s elevation axis and $o_{l}x_{l}$ represents the optical axis of the theodolite. Target coordinate system $S(x_{s}y_{s}z_{s})$: the origin is the target and $o_{s}x_{s}$ is parallel to $o_{l}x_{l}$.

Given a cartesian coordinate of the target in the S coordinate system as ***S*** = [0, 0, 0, 1]^T^, the pointing model of a VSRT can be derived using homogeneous coordinate transformation and is defined as follows:(1)$$G=T_{S}^{G}S=Rot(Z,A_{g})Rot(X,E_{g})Trans(d,0,0)S$$
(2)$$T_{S}^{G}=\left[\begin{array}{cccc} \cos(A_{g}) & \sin(A_{g}) & 0 & 0\\ -\sin(A_{g}) & \cos(A_{g}) & 0 & 0\\ 0 & 0 & 1 & 0\\ 0 & 0 & 0 & 1 \end{array}\right]\left[\begin{array}{cccc} 1 & 0 & 0 & 0\\ 0 & \cos(E_{g}) & \sin(E_{g}) & 0\\ 0 & -\sin(E_{g}) & \cos(E_{g}) & 0\\ 0 & 0 & 0 & 1 \end{array}\right]\left[\begin{array}{cccc} 1 & 0 & 0 & d\\ 0 & 1 & 0 & 0\\ 0 & 0 & 1 & 0\\ 0 & 0 & 0 & 1 \end{array}\right]$$
where *G* = [*x_g_*, *y_g_*, *z_g_*, 1]^T^, $T_{S}^{G}$ represents the homogeneous coordinate transformation matrix between coordinate systems from *S* to *G*. Here, Rot denotes the rotation matrix, Trans signifies the translation matrix, and *d* is the output value of ranging. In the context of coordinate transformation, positive and negative angles are defined as counterclockwise and clockwise rotations, respectively.

According to Formula (1), the theoretical azimuth and elevation pointing angles of the VSRT, $A_{g}$ and $E_{g}$, respectively, in ground plane coordinates can be determined.
(3)$$\left\{\begin{array}{l} A_{g}=\arccos\left[x_{g}/{\left(y_{g}^{2}+z_{g}^{2}\right)}^{1/2}\right]\\ E_{g}=\arcsin\left[z_{g}/{\left(x_{g}^{2}+y_{g}^{2}+z_{g}^{2}\right)}^{1/2}\right] \end{array}\right.$$

Due to inaccuracies in VSRT processing, assembly, and image processing, a deviation exists between the ideal pointing angle and the actual pointing angle, known as pointing error. This error can be further broken down into azimuth pointing error and elevation pointing error. The actual azimuth and elevation angles of the target in ground plane coordinates can be determined through a series of coordinate transformations. Subsequently, the azimuth pointing error, denoted by ΔA, and elevation pointing error, denoted by ΔE, are calculated as follows:(4)$$\left\{\begin{array}{l} \Delta A={\hat{A}}_{g}-A_{g}=f(\cdot )\\ \Delta E={\hat{E}}_{g}-E_{g}=g(\cdot ) \end{array}\right.$$
where ${\hat{A}}_{g}$ is the actual azimuth pointing angle; ${\hat{E}}_{g}$ is the actual elevation pointing angle; and f(⋅) and g(⋅), respectively, denote the error models for azimuth angle and elevation angle, specifically referring to the model for correcting pointing error.

The process of pointing error correction involves the following steps: (1) establishing an analysis and correction model for pointing error through an error analysis and homogeneous coordinate transformation, (2) determining the values of the VSRT error term and obtaining pointing error data based on the simulation results, (3) utilizing the regression analysis method to calculate the regression coefficient of the correction model, and (4) applying the derived model to correct the pointing error.

## 3. Analysis of Error Source and Identification of Error Mode

### 3.1. Source of Pointing Error

Errors impacting the pointing accuracy can arise at various stages of the VSRT’s lifecycle, including manufacturing, assembly, installation, and operation. The primary sources of error are illustrated in Figure 3 and encompass the mechanical tracking frame error, encoder angle error, image system error, single-photon ranging error, shaking error in the supporting platform, and orientation error. Among these error sources, as depicted in Figure 3, the position error of the orientation component has not been taken into account in subsequent modeling. This is due to the fact that the position error only amounts to a few centimeters. However, when compared to the significant imaging distance of the VSRT, which ranges approximately from tens to hundreds of kilometers, this discrepancy becomes negligible. As a result, it can be disregarded in subsequent modeling analysis.

### 3.2. Transformation of Coordinates for Error Source Identification

The error in the alignment of the LOS coordinate system *L* with respect to the target coordinate system *S*, as depicted in Figure 4a, primarily encompasses the single-photon ranging error, optical axis jitter and consistency error, and position errors in target graphics processing. The transformation matrix from *S* to *L* is as follows:(5)$$T_{S}^{L}=Rot(Z,\Delta \theta_{x})Rot(Y,\Delta \theta_{y})Trans(d+\Delta d,0,0)$$
where *d* + Δ*d* represents the single-photon ranging measurement and its associated error, while $\Delta \theta_{x}$ and $\Delta \theta_{y}$ are derived from the conversion of target miss quantities Δ*X* and Δ*Y* resulting from image processing.

The LOS coordinate system still exhibits jitter and consistency errors. The conversion matrix is as follows:(6)$$T_{L}^{L_{0}}=Rot(Z,\Delta \varphi_{zl})Rot(Y,\Delta \varphi_{yl})$$
where *L*_0_ is the ideal coordinate system of *L*, and $\Delta \varphi_{zl}$ and $\Delta \varphi_{yl}$ are the sum of the amount of jitter and consistency error of LOS around the *Z* and *Y* axes, respectively.

Due to the adoption of a two-axis frame, the VSRT exhibits no rotation error between LOS and the elevation shafting. Consequently, only the optical axis verticality error and errors between the LOS coordinate system *L* and elevation axis coordinate system *E* remain, as depicted in Figure 4b, with an associated error conversion matrix.
(7)$$T_{L_{0}}^{E}=Rot(Z,\Delta \varphi_{c3})Rot(Y,\Delta \varphi_{c2})$$

The error in shaft rotation encompasses both shaft jitter and encoder angle measurement inaccuracies, shown in Figure 5a. Set *E*_0_ as the ideal coordinate system of *E*. The transformation matrix for the motion of the shaft is presented as follows:(8)$$T_{E}^{E_{0}}=Rot(Y,-({\hat{E}}_{b}+\Delta \phi_{E}))Rot(Z,\Delta \varphi_{e3})Rot(X,\Delta \varphi_{e1})$$

The elevation shafting error comprises two types of errors, verticality error and rotation error, as depicted in Figure 5b. The verticality error is primarily attributed to the non-orthogonal sum of the elevation axis and azimuth axis, denoted by $\Delta i_{3}$ and $\Delta i_{1}$. The error transformation matrix is presented below:(9)$$T_{E_{0}}^{A}=Rot(X,\Delta i_{3})Rot(Z,\Delta i_{1})$$

The azimuth shafting error encompasses verticality, rotation, and orientation errors, as illustrated in Figure 6a. Set *A*_0_ as the ideal coordinate system of *A*. When the azimuth shafting undergoes rotation, the error conversion matrix is as follows:(10)$$T_{A}^{A_{0}}=Rot(Z,-({\hat{A}}_{b}+\Delta \phi_{A}))Rot(Y,\Delta \varphi_{a2})Rot(X,\Delta \varphi_{a1}).$$
where $\Delta \phi_{A}$ is the sum of the encoder angle measurement error and orientation error, and $\Delta \varphi_{a1}$ and $\Delta \varphi_{a2}$ are the jitter amount of the azimuth shafting around the *X* axis and the *Y* axis, respectively.

The vertical error in the azimuth shafting is attributed to the lack of perpendicularity between the azimuth shafting and the base plane, shown in Figure 6b. The corresponding transformation matrix is as follows:(11)$$T_{A_{0}}^{B}=Rot(Y,\Delta \varphi_{b12})Rot(X,\Delta \varphi_{b11})$$

The errors of the vehicle supporting platform primarily consist of shaking in the elevation and rolling directions, denoted by $\Delta \varphi_{b22}$ and $\Delta \varphi_{b21}$. The conversion matrix is as follows:(12)$$T_{B}^{G}=Rot(Y,\Delta \varphi_{b22})Rot(X,\Delta \varphi_{b21})$$

## 4. Piecewise Linear Regression Model

### 4.1. Construction of Piecewise Linear Regression Model

In conclusion, based on Equations (5)–(12), the actual location of the target in coordinate system *G* is determined by
(13)$$\overset{\wedge }{G}={\left[x_{g},y_{g},z_{g},1\right]}^{T}=T_{B}^{G}T_{A_{0}}^{B}T_{A}^{A_{0}}T_{E_{0}}^{A}T_{E}^{E_{0}}T_{L_{0}}^{E}T_{L}^{L_{0}}T_{S}^{L}S.$$

When simplifying the above equation, it is assumed that each error factor Δ is infinitesimal, resulting in cosΔ=1 and sinΔ=0, while disregarding the second-order and higher-order error terms. Formula (13) can be further simplified as follows:(14)$$\left\{\begin{array}{l} x_{g}=(d+\Delta d)(\cos(A_{g})\cos(E_{g})+(\Delta \varphi_{c3}+\Delta \theta_{x}+\Delta \varphi_{l3})\sin(A_{g})+\Delta \varphi_{b2}\sin(E_{g})\\ +(\Delta \phi_{A}+\Delta \varphi_{e3}+\Delta \varphi_{a1}+\Delta \varphi_{a2})\sin(A_{g})\cos(E_{g})+(\Delta \phi_{E}+\Delta \theta_{y}+\Delta \varphi_{l2})\cos(A_{g})\sin(E_{g})\\ +(\Delta \varphi_{e1}+\Delta i_{3})\sin(A_{g})\sin(E_{g}))\\ y_{g}=(d+\Delta d)(\sin(A_{g})\cos(E_{g})-(\Delta \varphi_{c3}+\Delta \theta_{x}+\Delta \varphi_{l3})\cos(A_{g})-\Delta \varphi_{b1}\sin(E_{g})\\ -(\Delta \phi_{A}+\Delta \varphi_{e3}+\Delta \varphi_{a1}+\Delta \varphi_{a2})\cos(A_{g})\cos(E_{g})+(\Delta \phi_{E}+\Delta \theta_{y}+\Delta \varphi_{l2})\sin(A_{g})\sin(E_{g})\\ -(\Delta \varphi_{e1}+\Delta i_{3})\cos(A_{g})\sin(E_{g}))\\ z_{g}=(d+\Delta d)((\Delta \phi_{E}+\Delta \theta_{y}+\Delta \varphi_{l2})\cos(E_{g})-\sin(E_{g})+\Delta \varphi_{b2}\cos(A_{g})\cos(E_{g})\\ -\Delta \varphi_{b1}\sin(A_{g})\cos(E_{g})) \end{array}\right.$$

According to the transformation relationship between spherical coordinates and rectangular coordinates, it is observed that when the azimuth angle is $A\in [0,90^{{}^{\circ}})$, there exists
(15)$$\left\{\begin{array}{l} \tan(A+\Delta A)=-\frac{y_{g}}{x_{g}}\\ \sin(E+\Delta E)=\frac{z_{g}}{d+\Delta d} \end{array}\right..$$

The azimuth range spans from 0 to 360° and is divided into three intervals. By incorporating Equation (14) into Equation (15), the piecewise linear function expression of the pointing error can be derived through the principle of model simplification. 

When $A\in [0^{{}^{\circ}},90^{{}^{\circ}})\&(270^{{}^{\circ}},360^{{}^{\circ}})$, we can obtain
(16)$$\left\{\begin{array}{l} \Delta A_{1,4}=(\Delta \phi_{A}+\Delta \varphi_{e3}+\Delta \varphi_{a1}+\Delta \varphi_{a2})+(\Delta \varphi_{e1}+\Delta i_{3})\tan E\\ +(\Delta \varphi_{c3}+\Delta \theta_{x}+\Delta \varphi_{l3})\sec E+\Delta \varphi_{b2}\sin A\tan E+\Delta \varphi_{b1}\cos A\tan E\\ \Delta E_{1,4}=(\Delta \phi_{E}+\Delta \theta_{y}+\Delta \varphi_{c2})+\Delta \varphi_{b2}\cos A-\Delta \varphi_{b1}\sin A \end{array}\right..$$

When $A\in [90^{{}^{\circ}},270^{{}^{\circ}}]$, we can obtain
(17)$$\left\{\begin{array}{l} \Delta A_{2,3}=-\Delta A_{1,4}\\ \Delta E_{2,3}=\Delta E_{1,4} \end{array}\right..$$

Based on the above expression, a specific functional relationship between the pointing error and each component error factor can be derived. Equation (16) is formulated as a multiple linear regression equation in the following manner:(18)y=Xβ+ε.
where $y={\left[\Delta A,\Delta E\right]}^{T}$ represents the dependent variables; $\beta ={\left[\beta_{0},\beta_{1},\ldots ,\beta_{7}\right]}^{T}$ denotes the regression coefficients; $\varepsilon ={\left[\varepsilon_{A},\varepsilon_{E}\right]}^{T}$ signifies the random error term; $\varepsilon \sim N(0,\sigma^{2}I_{n})$; and X refers to the regression design matrix encompassing the independent variable. Its mathematical expression is as follows:(19)$$X={\left[\begin{array}{cc} 1 & 0\\ \tan E & 0\\ \sec E & 0\\ \sin A\tan E & 0\\ \cos A\tan E & 0\\ 0 & 1\\ 0 & \cos A\\ 0 & -\sin A \end{array}\right]}^{T}$$

Hence, the model for the piecewise linear regression of the pointing error in a VSRT is formulated.

### 4.2. Calculation of Regression Coefficients 

The unknown coefficients of the multiple linear regression equation are estimated using the least squares method. The goal is to minimize the sum of squares of deviations in the regression model y=Xβ+ε, ultimately solving for $\min\left\{{\left(y-X\beta \right)}^{T}(y-X\beta )\right\}$ to obtain the final estimate of the regression coefficients.
(20)$$\overset{\wedge }{\beta }={\left(X^{T}X\right)}^{-1}X^{T}y$$

### 4.3. Evaluation of Regression Model

In order to establish the relationship between the dependent variable ***y*** and the independent variable ***X***, it is imperative to assess the regression equation using statistical methods. It is essential to make an assumption of normality.

Significance testing is conducted using the *F*-test to directly assess the significance of the regression equation based on the decomposition of sum of squares. The sum of squares is decomposed as follows:
*SST* = *SSR* + *SSE*(21)
where *SST* represents the total sum of squares; *SSR* signifies the regression sum of squares; and *SSE* refers to the error sum of squares.

The F-test statistic, denoted by *F*, is calculated as follows:(22)$$F=\frac{SSR/p}{SSE/(n-p-1)}$$
where *p* represents the number of explanatory variables, while *n* denotes the number of observations in the multiple linear regression equation. The distribution of the statistic follows the *F* distribution with degrees of freedom (*p*, *n* − *p* − 1).

The goodness of fit test assesses the degree to which the regression equation fits the sample observations. In multiple linear regression, this is evaluated using a statistic, ${\bar{R}}^{2}$, known as the adjusted coefficient of determination, which is defined as follows:(23)$${\bar{R}}^{2}=1-\frac{\frac{SSE}{n-p-1}}{\frac{SST}{n-1}}$$

The degrees of freedom for *SSE* and *SST* are represented as *n* − *p* − 1 and *n* − 1, respectively. The sample adjusted determination coefficient ${\bar{R}}^{2}$ falls within the range of 0 to 1. Its value is closer to 1, indicating a more favorable regression fitting effect. While ${\bar{R}}^{2}$ offers a clearer and more intuitive reflection of the regression fitting effect compared to the *F*-test, it should not be solely relied upon as a strict significance test. Therefore, a comprehensive evaluation of the quality of the regression model can be achieved through the combination of both F-test and goodness of fit test methods.

## 5. Experiment and Results

### 5.1. Data Acquisition and Model Solving

Three approaches are usually available for acquiring VSRT regression analysis data: the outdoor shooting calibration stars method, the indoor utilization of a specialized detection device method, and the simulation calculation method based on structural parameters. The first two methods involve collecting multiple sets of data, which is time-consuming and labor-intensive. In this study, a VSRT with an optical aperture (Manufacturer: Changchun Institute of Optics, Fine Mechanics and Physics, Chinese Academy of Sciences, Changchun, China) of 450 mm was used. Sixteen actual error parameters were successfully obtained through appropriate process installation and detection methods during the installation of the VSRT, as presented in Table 1. These error parameters were substituted into Formula (1) and Formula (13), where *A* = 0~360° is evenly divided into 60 parts to obtain *A**_i_***, i=1,…,60, and *E* = 0~60° is equally divided into 60 parts to obtain *B**_j_***, j=1,…,60; any combination of these can yield 60 × 60 sets of observation data. As depicted in Figure 7, the curve illustrates the variation in pointing error in relation to azimuth and elevation angle. Azimuth angle distinct discontinuity is evident at *A* = 90° and 270°. Consequently, it is imperative to employ a piecewise linear regression model for accurately characterizing pointing error across the entire operational range of the VSRT.

To derive the linear regression equation, we substituted the aforementioned data into Equation (20) and computed the regression coefficients using the least squares method. To mitigate the impact of multicollinearity on the model, we employed a stepwise regression optimization approach to determine the regression coefficients.
(24)$$\beta ={\left[-2.304,-3.414,-6.233,-5.001,-5.000,-4.033,-5.000,5.000\right]}^{T}$$

According to the normality hypothesis, the model underwent significance of *F*-testing and goodness of fit evaluation. The results indicated that the regression equation and regression coefficients were statistically significant, and the fitting effect was deemed satisfactory. Figure 8 displays the histogram of the standardized residual distribution for azimuth error and elevation error estimates, revealing a predominantly normal distribution around the baseline.

The change curve of the VSRT pointing error, estimated by the piecewise linear regression model, is depicted in Figure 9. It is evident that the piecewise linear regression model exhibits characteristics of parsimony and clear physical interpretation, enabling rapid estimation of pointing errors for various azimuth and elevation angles. This lays a solid foundation for subsequent correction of pointing error.

### 5.2. Experiment and Results

The experimental setup, as depicted in Figure 10, is used by the 450 mm optical aperture VSRT. The experimental setup, as depicted in Figure 10, consists of a VSRT with an optical aperture of 450 mm. This VSRT is equipped with both a visible light imaging system and a single-photon ranging system, and it is mounted on a vehicle supporting platform alongside a collimator target and data acquisition/display system. The VSRT, which has a focal length of 2000 mm, was securely affixed to the platform during the experiment, wherein it underwent sinusoidal motion to track the collimator target at a predetermined angular velocity. The data acquisition system records the real-time azimuth and elevation angles of the VSRT, along with their corresponding timestamps, while simultaneously capturing video images. The data interpretation system is employed to analyze deviations in azimuth and elevation for the collimator target image. By dividing the deviation distance by the focal length of the VSRT, the actual pointing error can be obtained.

To assess the efficacy of the correction model in reducing pointing errors, four dynamic tracking experiments were conducted using the VSRT. Test data for each experiment were obtained through multiple measurements. Figure 11 illustrates the azimuth and elevation pointing errors, as well as the corrected scatter plot. It is evident that the maximum pointing error post-correction decreased from 13.2″ to 4.9″, representing a reduction of 62.9%. 

The comprehensive pointing error of the VSRT is quantified by its root mean square (RMS) [26], as a measure of accuracy.
(25)$$\Delta =\sqrt{\Delta A^{2}+\Delta E^{2}}$$

The comprehensive pointing error correction effect of the VSRT is depicted in Figure 12. It is evident that, following the correction, the overall pointing error is reduced to 4.4%~35.5% of its original value, and the RMS value of pointing error after correction decreases from a maximum of 17.2″ to less than 5″, achieving arc-second accuracy. 

## 6. Discussion

The VSRT’s operational range spans from 0 to 360° in azimuth and from 0 to 60° in elevation. Given any combination of azimuth and elevation angles within this range, the proposed model in this paper is capable of swiftly and accurately estimating the pointing error under these conditions. Additionally, the model is characterized by its low computational complexity and simplicity. It is suitable not only for post-correction of VSRT pointing error but also for real-time correction with minimal system resources. The model also exhibits high practicality and applicability in various theodolites.

After correction for the VSRT, the test results of its pointing error indicate that the residual error in the first to fourth tests is very small, all below 2″. However, in the 5th to 16th tests, the residual error ranged between 2″ and 5″ and showed no regularity. This suggests that nonlinear factors have minimal impact on the pointing error. Nonetheless, there are still some residual errors attributed mainly to non-ideal rigidity of the vehicle supporting platform. As the attitude changes during dynamic tracking of targets by VSRT, varying levels of shaking are produced by the platform. This amount of shaking is treated as a constant in our correction model. Therefore, it is crucial to focus on researching and devising a correction method for VSRT pointing error that is applicable to the sloshing dynamics of a non-ideally rigid platform. We anticipate that the non-linear error will be corrected based on model modification, and appropriate methodologies will be implemented to mitigate platform shaking error. It is expected that the pointing accuracy of the VSRT will undergo further improvement.

## 7. Conclusions

This paper presents a novel method for correcting the pointing error of a VSRT and conducts an in-depth analysis of the sources contributing to the pointing error. We establish a comprehensive pointing error model for the VSRT and innovatively divide the entire process into two linear regression models, followed by statistical evaluation. Additionally, we design and implement an experimental setup to assess the pointing error of a VSRT with a 450 mm optical aperture. The results demonstrate a remarkable correction effect, reducing the RMS value from 17″ to less than 5″, achieving accuracy within arc-seconds. The proposed correction method in this study has substantial potential to enhance the pointing accuracy of a VSRT and to provide essential technical support for improving its high-performance capabilities.

## Figures and Tables

**Figure 1 sensors-24-03192-f001:**
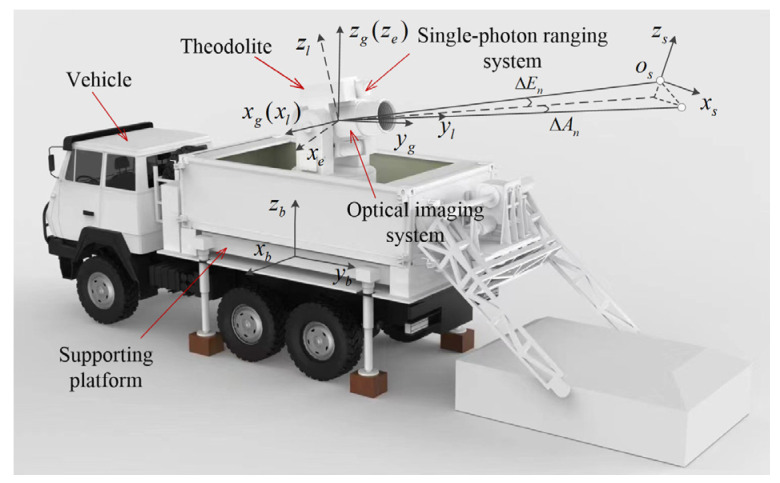
Structure of VSRT.

**Figure 2 sensors-24-03192-f002:**
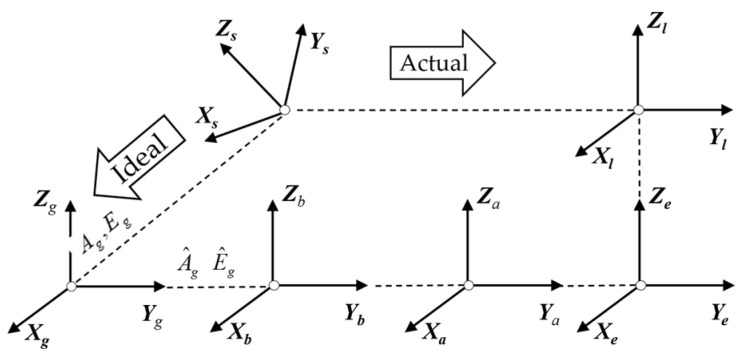
Process of target pointing in VSRT.

**Figure 3 sensors-24-03192-f003:**
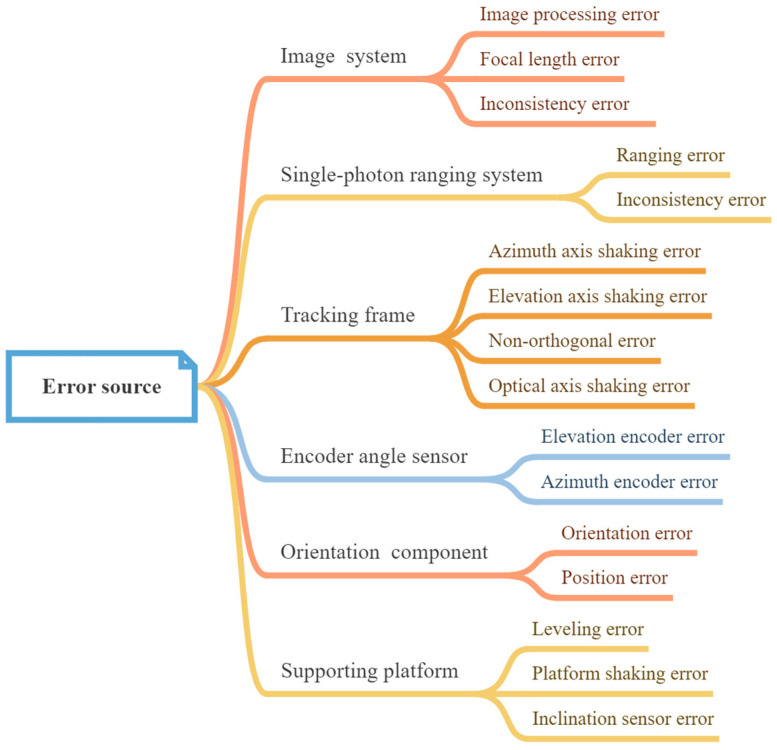
Error sources.

**Figure 4 sensors-24-03192-f004:**
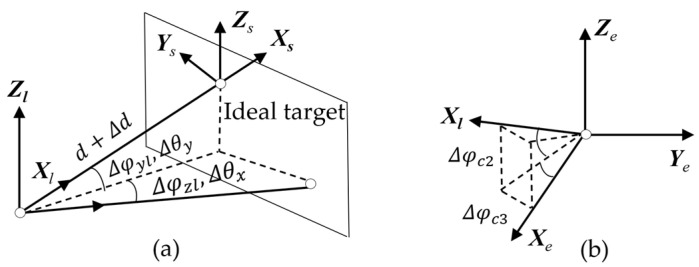
LOS shafting error. (**a**) Position error of imaging target and (**b**) perpendicularity error of LOS axis.

**Figure 5 sensors-24-03192-f005:**
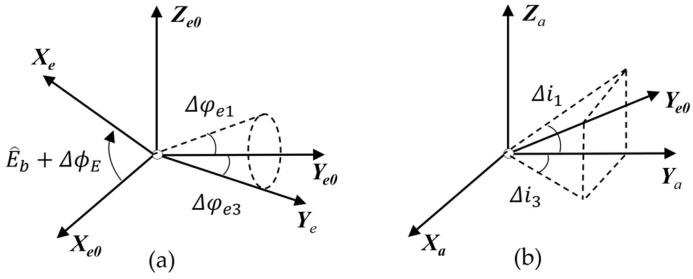
(**a**) Rotation and encode error and (**b**) perpendicularity error of elevation axis.

**Figure 6 sensors-24-03192-f006:**
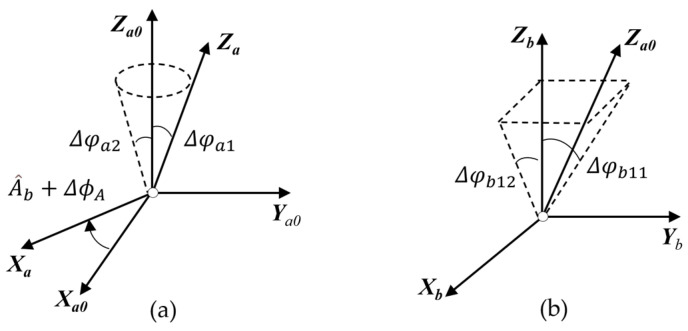
(**a**) Rotation and encode error and (**b**) perpendicularity error of azimuth axis.

**Figure 7 sensors-24-03192-f007:**
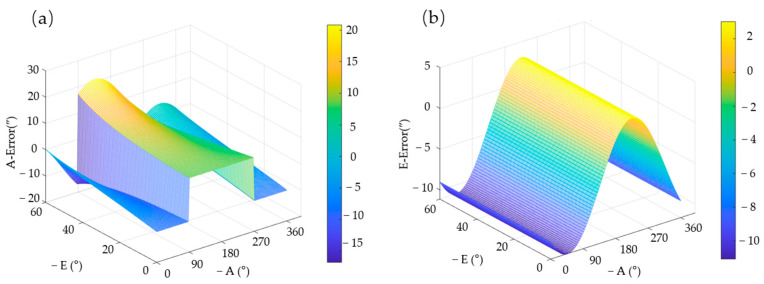
The variation in pointing error in relation to (**a**) azimuth angle and (**b**) elevation angle.

**Figure 8 sensors-24-03192-f008:**
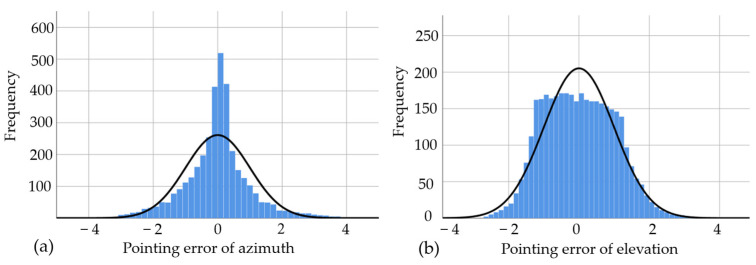
The normalized residual distribution histograms obtained from the regression analysis: (**a**) pointing error of azimuth and (**b**) pointing error of azimuth.

**Figure 9 sensors-24-03192-f009:**
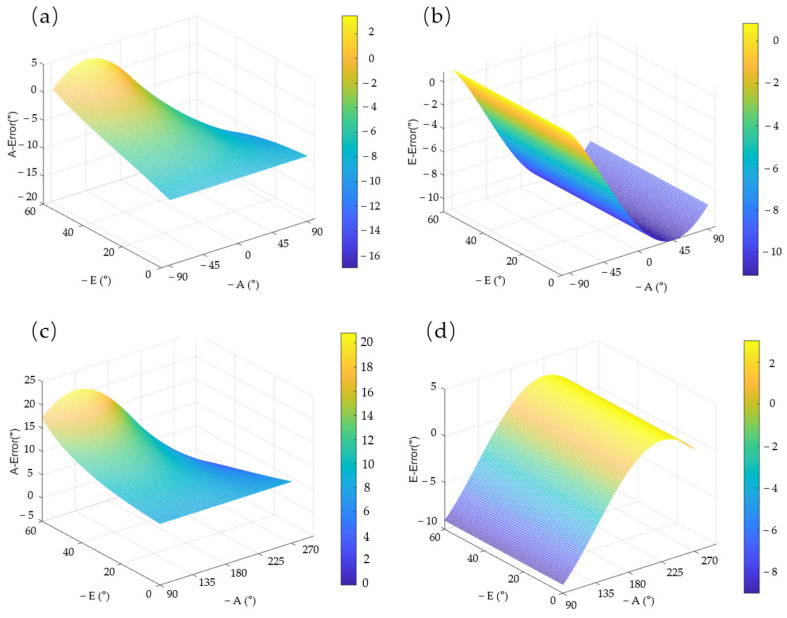
The estimated pointing error varies with azimuth and elevation angles. (**a**) The azimuth pointing error at the azimuth angle from −90° to 90°; (**b**) the elevation pointing error at the azimuth angle from −90° to 90°; (**c**) the azimuth pointing error at the azimuth angle from 90° to 270°; (**d**) the elevation pointing error at the azimuth angle from 90° to 270°.

**Figure 10 sensors-24-03192-f010:**
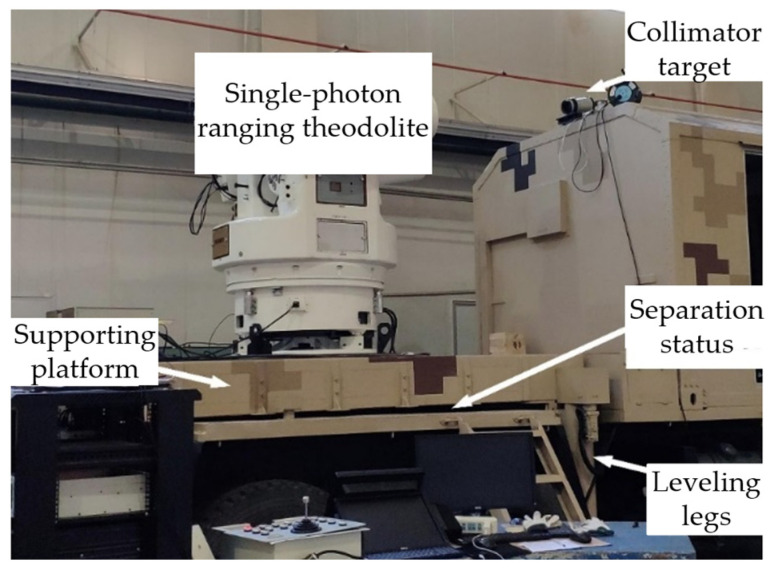
Experiment setup.

**Figure 11 sensors-24-03192-f011:**
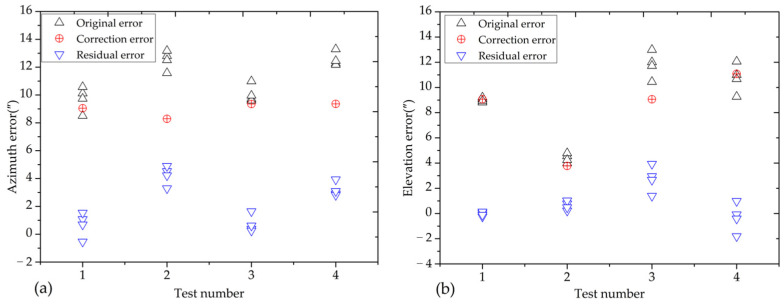
Scatter plot diagram of pointing errors in (**a**) azimuth and (**b**) elevation.

**Figure 12 sensors-24-03192-f012:**
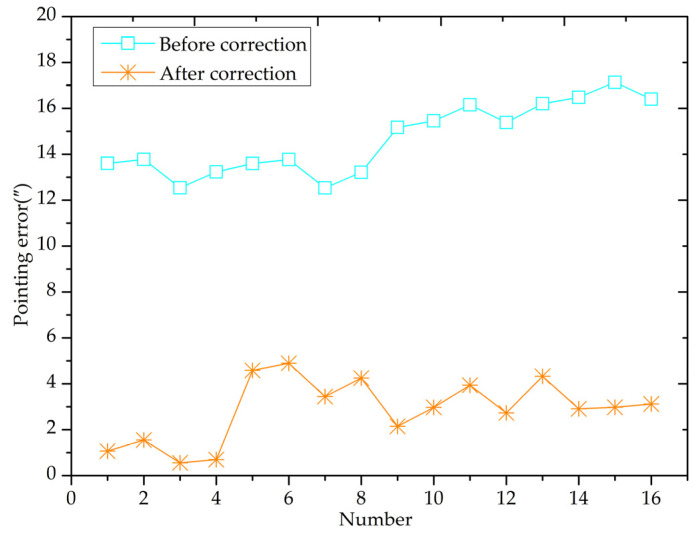
Diagram of comprehensive pointing error after correction.

**Table 1 sensors-24-03192-t001:** Error identification used in pointing model.

Order	Error Description	Error Symbol
1	Inclination error of supporting platform around $o_{g}x_{g}$	$\Delta \varphi_{b1}$
2	Inclination error of supporting platform around $o_{g}y_{g}$	$\Delta \varphi_{b2}$
3	Encode angle error of azimuth axis	$\Delta \phi_{A}$
4	Encode angle error of elevation axis	$\Delta \phi_{E}$
5	Shaking error of azimuth axis around $o_{b}x_{b}$	$\Delta \varphi_{a1}$
6	Shaking error of azimuth axis around $o_{b}y_{b}$	$\Delta \varphi_{a2}$
7	Non-orthogonal error between elevation and azimuth axes	$\Delta i_{1}$
8	Non-orthogonal error between elevation and azimuth axes	$\Delta i_{3}$
9	Shaking error of the elevation axis around $o_{e}x_{e}$	$\Delta \varphi_{e1}$
10	Shaking error of the elevation axis around $o_{e}z_{e}$	$\Delta \varphi_{e3}$
11	Non-orthogonal error between optical axis and elevation axis	$\Delta c_{2}$
12	Non-orthogonal error between optical axis and elevation axis	$\Delta c_{3}$
13	Horizontal jitter error of optical axis	$\Delta \varphi_{l2}$
14	Vertical jitter error of optical axis	$\Delta \varphi_{l3}$
15	Position error of target image in x direction of image plane	$\theta_{\Delta x}$
16	Position error of target image in y direction of image plane	$\theta_{\Delta y}$

## Data Availability

The data supporting the findings in this paper are currently not accessible to the public; however, they can be obtained from the authors upon reasonable request.

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
