# Peer review of "Pointing Error Correction for Vehicle-Mounted Single-Photon Ranging Theodolite Using a Piecewise Linear Regression Model"

_sensors, 2024, doi:10.3390/s24103192_

Round 1

Reviewer 1 Report

Comments and Suggestions for Authors

The paper proposes a pointing error correction method based on a piecewise linear regression model for VSRT. From the result, it can be shown point error is reduced to almost arc second level just using pure mathematical model, which is impressive. The paper is innovative, with sufficient theoretical analysis and experimental testifying. However, because of the bad quality of figures in manuscript, some details are hard to read or understand, therefore, before been accepted, serval question need to be answered:

1.      Quality of all figures are quite low, some figures are hardly to read, please improve according to the rules of journal.

2.      In section 3.1, 16 different pointing error source for VSRT are listed, because of the bad quality of the figure, I am not sure if each error source is identified in subsequent model, please clarify each error source model.

3.      In section 5.1, authors mentioned outdoor calibration could be used for data obtaining, from my opinion, it is easy to complete this step with experiment setup in 5.2 figure 10,

4.      Only focal length and aperture size of experiment setup is mentioned in 5.2, how to evaluate the correctness and suitability for experiment setup compare with data acquisition method based on simulation in 5.1?

5.      In section 5.2, the method of obtaining point error is unclear, could authors provide it in details, which I think is quite important for efficacy evaluation.

6.      In general, time consumption of proposed method is not mentioned, is it possible to apply proposed model in real time?

7.      For a future work, nonlinear error source could be corrected based on model modification, could authors provide a prospect on this?

Reviewer 2 Report

Comments and Suggestions for Authors

The manuscript provides a comprehensive analysis of the sources contributing to the pointing error of the VSRT and proposes a novel method for correction. The technical content is thorough and well-structured, with clear explanations of the theoretical framework, experimental setup, and results interpretation. The piecewise linear regression model developed for characterizing pointing error demonstrates a sound mathematical basis and is effectively applied to experimental data, leading to a significant reduction in pointing error. The discussion of residual errors and their implications adds depth to the analysis, while the conclusion effectively summarizes the key findings and highlights the potential impact of the proposed correction method. Additionally, providing insights into the limitations of the study and suggestions for future research directions would strengthen the overall contribution of the manuscript to the field of VSRT technology. Overall, the manuscript represents a valuable contribution to the pointing error correction methods for VSRT systems. However, the presentation quality of the manuscript is poor, thus the present version is not suitable for publication.

1. Please also define the abbreviation “RMS” in the Abstract.

2. All the figures should be improved. Each figure in the present manuscript is blurry. Moreover, please provide more detailed legends.

3. Please include more detailed discussions regarding the piecewise linear regression model's practicability in estimating pointing errors for different azimuth and elevation angles.

4. Please provide more details about the experimental setup to improve the repeatability.

5. Please further improve the manuscript formatting such as the unnumbered equation in Line 310.

Comments on the Quality of English Language

Moderate editing of English language required

Round 2

Reviewer 2 Report

Comments and Suggestions for Authors

Thanks for the responses. However, unfortunately, the figures in the PDF version are still blurry.
